# Gas Turbine Intercoolers: Introducing Nanofluids—A Mini-Review

**Ali Alsayegh** [1,*] **and Naser Ali** [2]

1    School of Aerospace, Transport and Manufacturing (SATM), Cranfield University, Cranfield MK43 0AL, UK
2    Nanotechnology and Advanced Materials Program, Energy and Building Research Center,
     Kuwait Institute for Scientific Research, Safat 13109, Kuwait; nmali@kisr.edu.kw
\*    Correspondence: a.alsayegh@cranfield.ac.uk

**Abstract:** Coolant is one of the main factors affecting the overall thermal performance of the intercooler for the gas turbine intercooled cycle. The thermal conductivity of conventional coolants, such as water, is relatively low when compared to solid conducting materials, and therefore can hinder the progress towards achieving a compact and highly effective intercooler. Nanofluids are advanced types of working fluids that contain dispersed nanoparticles in conventional basefluids, and as such possess superior thermal conductivity compared to their counterparts. In this paper, a short review on the effect of different nanofluids on the thermal performance of gas turbines intercoolers is presented for the first time. Firstly, this work reviews the different designs of intercoolers used in gas turbines intercooled cycles. Then, it explains the different types of nanofluids and their fabrication processes. The effective parameters, such as physical stability, thermal conductivity, and viscosity are also highlighted and discussed. Furthermore, the level of enhancement in the performance of intercoolers utilizing nanofluids is demonstrated and evaluated. Lastly, the current challenges and future research directions in this field are provided.

**Keywords:** heat exchanger; suspension; stability; thermophysical properties; working fluid

---

## 1. Introduction

With the continuous increase in energy demands and fluctuation in fuel prices, developers in the gas turbine industry have been working intensively into enhancing the performance of their machines to the highest possible limits that the currently existing technology can handle [1]. In general, if a manufacturer was to design and develop a new gas turbine with extensively modified features, the time that would be required from the initial conceptual phase to the production stages would typically take about ten or more years to complete. A more feasible and less time-consuming approach is to partially advance pre-existing designs by introducing intercooled, recuperated, and reheated cycles to the system. An example of such approach was demonstrated by Westinghouse and Rolls-Royce through their WR-21 marine gas turbine engine, where both intercooler and recuperator devices were used to improve the thermal performance of the system, and hence raise the efficiency of the engine [1]. Moreover, as these devices (i.e., reheater, intercooler, and recuperator) are forms of heat exchangers (HEs), they can be found in a variety of constructional designs, such as plate HEs (PHEs), plate-fin HEs (PFHEs), and plate-and-frame HEs (P&FHEs); flow arrangements (i.e., counterflow, parallel flow, and crossflow); and pass arrangements (i.e., multi- and single-pass) [2]. Selecting the appropriate type of HE depends on the application for which the thermal transport device is intended. Furthermore, many approaches have been used by scholars and designers to enhance the thermal efficiency of these equipment. Nevertheless, scientists have come to a point where no significant improvement can be achieved through only relying on the modification of the design itself, and thus the focus should

be redirected towards researching how to improve the thermal properties of pre-existing working fluids [3]. This is when, in 1993, Masuda et al. [4] proposed the use of ultra-fine metallic dispersions that were later on known as 'Nanofluids', as defined by Choi and Eastman [5] in 1995. These suspensions made of a mixture of nanoscaled particles and basefluid(/s) possess order of magnitude higher, effective thermal conductivity compared to their counterparts, and hence would have a large potential to improve the performance of HE devices, such as the intercooler, in a gas turbine cycle. The downside of employing nanofluids is that these advanced fluids have higher effective viscosity than their basefluids and tend to be physically unstable. In addition, these types of working fluids have yet to be explored in real-life intercoolers, which is the main focus of this work.

A survey on the available published work on nanofluids' employment in gas turbines intercoolers was performed through the most famous scientific search engines [6], namely Elsevier's abstract and citation database, Scopus; Clarivate Analytics abstract and citation database, Web of Science; and ProQuest abstract database, CSA Illumina. It was found that the Scopus database had the largest coverage on the topic, where 27 documents (2012–2020) [7] and 5 documents (2015–2016) [8] were seen to contain the terms 'Gas turbine nanofluid' and 'Intercooler nanofluid', respectively in their title, abstract, and/or keywords. Figure 1 illustrates the results found from the three search engines.

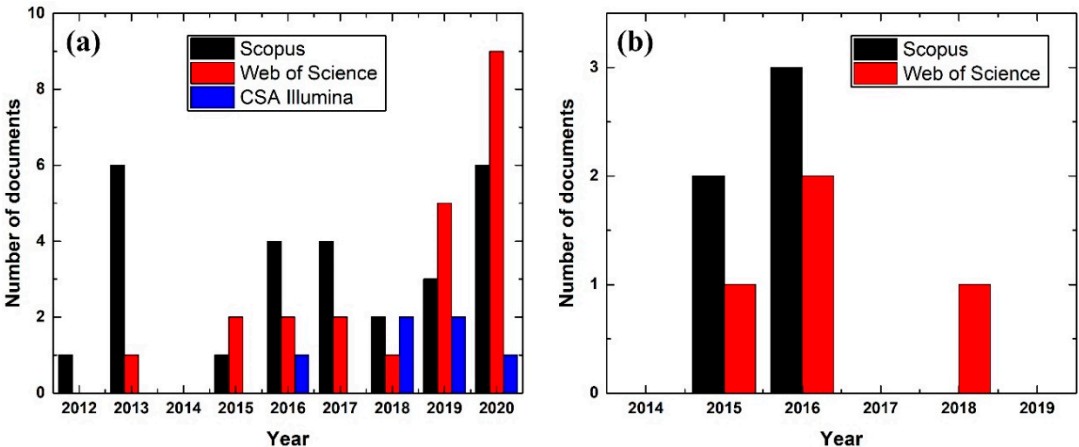

**Figure 1.** Search results obtained from Scopus, Web of Science, and CSA Illumina databases, where (**a**) shows the number of documents that were published from 2012–2020 with the term 'Gas turbine nanofluid' [7,9,10], and (**b**) demonstrates those containing the phrase 'Intercooler nanofluid' [8,11].

It is important to note that the CSA Illumina search engine did not show any results for the term 'Intercooler nanofluid', while the Google Scholar search engine was excluded from the investigation because most of the results were irrelevant to the targeted topic. Nevertheless, the low amount of published documents that were found in the conducted survey can indicate the following possibilities:

1.  The gas turbine industry has not yet opened up to this sort of advanced working fluids;
2.  The researchers working on the nanofluid field may have preferred to use the term HE to refer to intercoolers, as there are 1313 documents (1996–2020) on 'Heat exchanger nanofluid' in the Scopus database;
3.  The term 'Nanofluid' is often replaced by other terms, such 'Suspension', 'Dispersion', and 'Mixture'; and/or
4.  Other publications do exist but are not covered by the explored search engines.

In this content, the in-hand short review will provide an overview on some of the available literature on gas turbine intercoolers and nanofluids, as their potential working media. To the best of the authors' knowledge, there does not exist any other similar work across the available literature, and as such, this review article is the first of its kind in the field. Section 2 starts by presenting the gas turbines intercooled system and their different designs. Then, Section 3 demonstrates the

different types of nanofluids and production routes, after which Section 4 highlights the nanofluid's stability and other effective parameters, namely density, specific heat capacity, thermal conductivity, and viscosity. Furthermore, Section 5 summarizes some of the published studies on intercoolers' performance enhancement from utilizing different types of nanofluids. Finally, Section 6 presents the current gaps in scientific knowledge that researchers need to tackle before such types of advanced working fluids can be commercially acceptable in the gas turbines industry.

## 2. Gas Turbines Intercooled System and Their Designs

A gas turbine is an engine that provides mechanical work as a result of the rotational movement of its blades which is caused by the continuous flow of the compressed gas [12]. The working fluid is accelerated within the system by the combustion stage in the cycle. Furthermore, the mechanical output can then be utilized directly or converted into electric power when the turbine is coupled with an electrical generator. Some gas turbines are designed to operate with high pressure ratios, in such cases, an intercooler, which is basically a heat exchanger, is added to these systems between the high-pressure and low-pressure compressors to reduce the compression work by lowering the temperature of the working fluid [13]. This approach (i.e., adding intercooler), if implemented in a gas turbine cycle that contains a regenerator unit, would result in less fuel consumption and lower energy demand by the compression process, and hence an increase in the net power output of the engine [14]. Therefore, it is seen as a useful method for rising the overall efficiency of a gas turbine cycle as well as lowering the emissions [15]. However, whenever an intercooled HE is employed to a gas turbine cycle, a reheating unit needs to be installed after the compressor to compensate for the loss in flow temperature. Thus, the overall cycle would consequently increase in size, along with its complexity [16]. Furthermore, gas turbines with intercooled systems can be found in aero-engines (e.g., General Electric LMS 100) [17], mid-land engines (e.g., Siemens SGT-A65) [18], and marine engines (e.g., Westinghouse and Rolls-Royce WR-21) [19]. Figure 2 shows the gas turbine engines of the three previous examples.

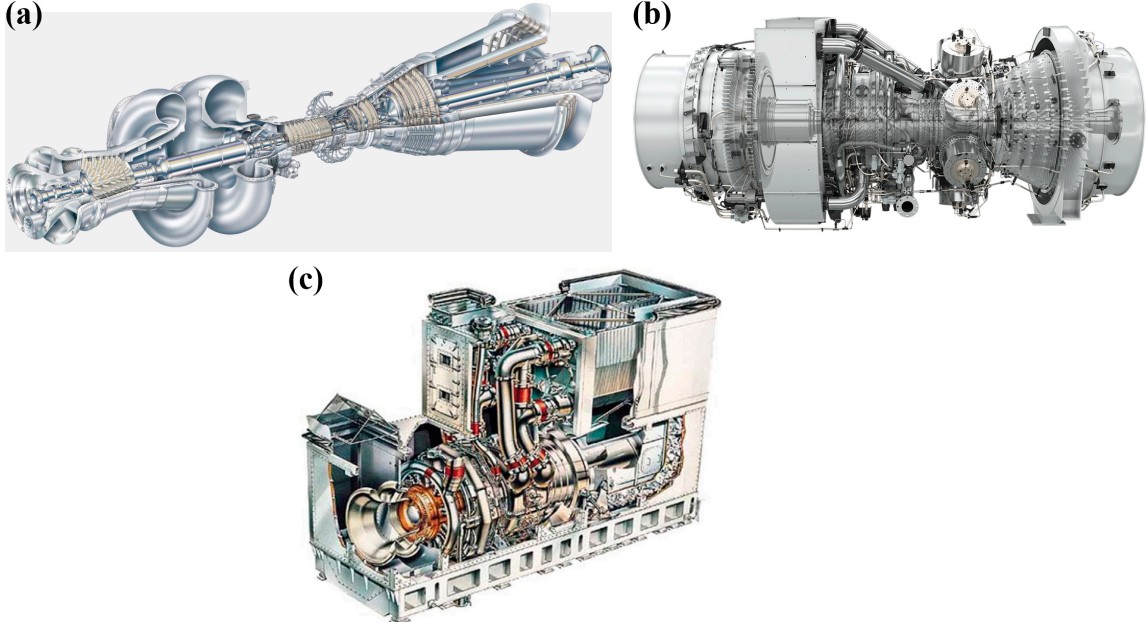

**Figure 2.** Gas turbine engines containing intercoolers, where (**a**) shows the General Electric LMS 100 engine, (**b**) demonstrates the Siemens SGT-A65 engine, and (**c**) illustrates the Westinghouse and Rolls-Royce WR-21 engine [17,18].

Moreover, the working fluids that pass through the intercooler device can be of gas–gas or gas–liquid, and the flow arrangements can be of counterflow, parallel flow, or crossflow [20]. In the counterflow arrangement, the two working fluids enter the system in opposite directions, while the

two flows move in the same direction in the parallel flow arrangement. In addition, the two working fluid flows move in a perpendicular manner in the crossflow arrangement. Figure 3 demonstrates the three flow arrangements. These different flow arrangements are used to enhance the performance of the thermal exchange device. Nevertheless, the thermophysical properties of the working fluid play a major role when it comes to the effectiveness of the intercooler unit, which is the case with all HEs [2].

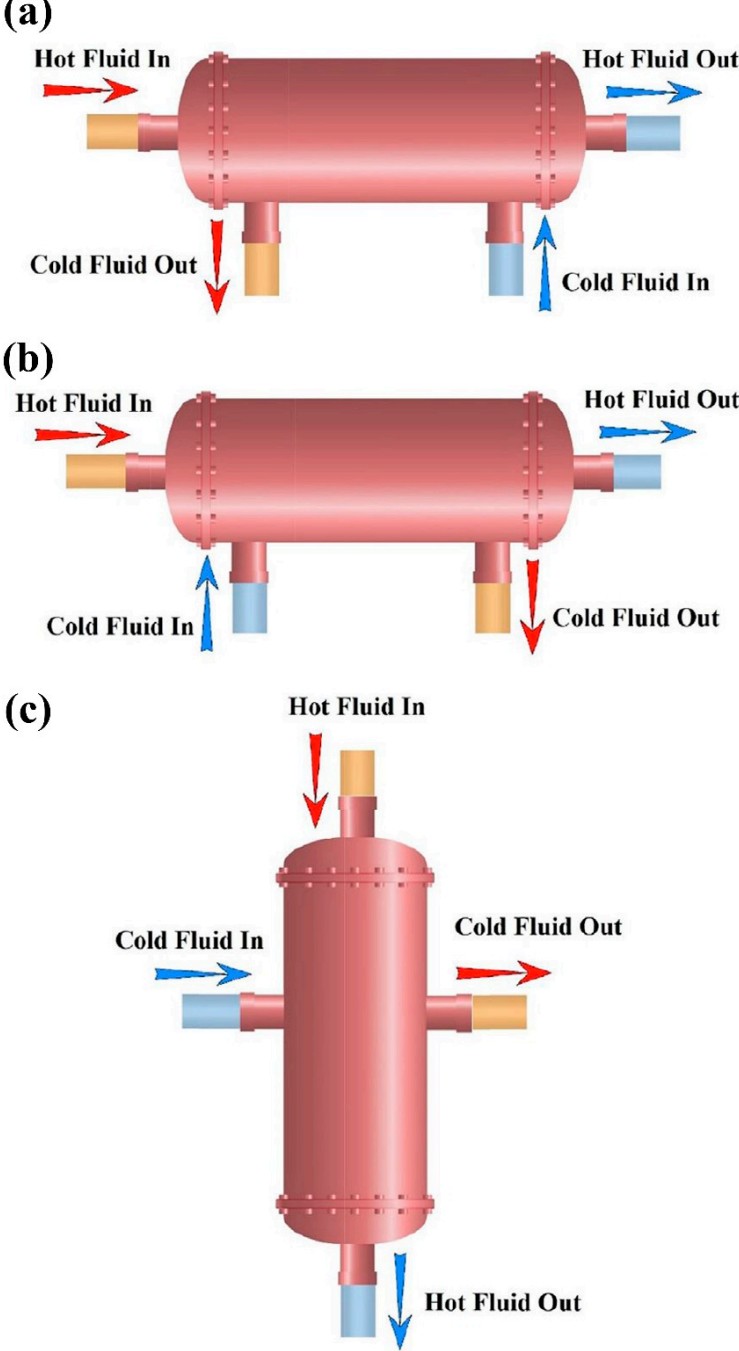

**Figure 3.** Different types of flow arrangements in intercoolers, where (**a**–**c**) shows the counterflow, parallel flow, and crossflow flows arrangements, respectively. Reproduced with permission from [21]. Elsevier, 2019.

In terms of the heat exchange design of the intercoolers, they commonly exist as shell and tube HEs or tube-fin HEs [20]. This is because these types of HEs meet the requirements of the working condition of the gas turbine intercooler. An example of the two previous HEs type are illustrated in Figure 4 [22].

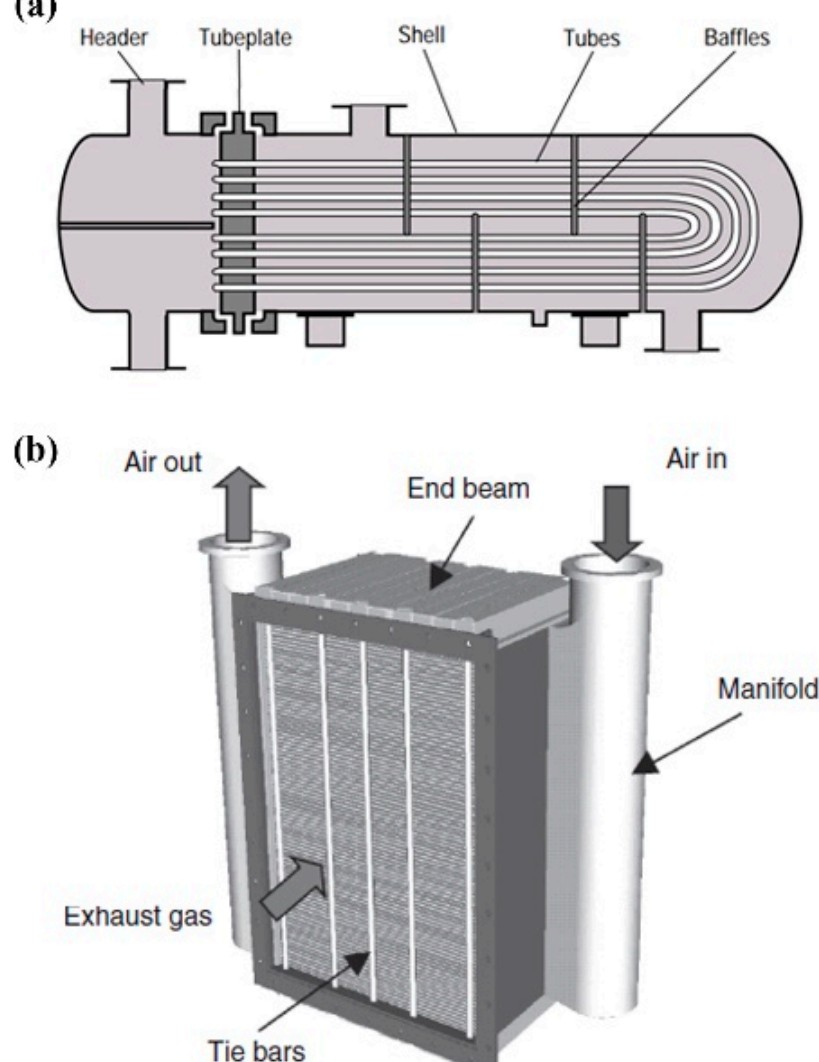

**Figure 4.** The two different intercoolers designs, where (**a**) illustrates a shell and tube heat exchanger, and (**b**) shows the tube-fin heat exchanger design [22].

There are many published works across the literature that demonstrates the improvements gained from employing intercoolers. For example, Wang and Pan [23] investigated the effect caused to the performance of a supercharged gas turbine plant from changing the location of the intercooler in an intercooled-supercharged gas turbine Brayton cycle. Their attempt caused the peak air pressure entering the combustion chamber of the gas turbine to increase, and thus achieving a maximum cycle pressure and thermal efficiency. The output of their research showed 20–30% improvement in the thermal efficiency and mass specific power over the conventional cycle. Guo et al. [24] experimentally studied the irreversibility of an intercooled Brayton gas turbine cycle, where they explored the effect of compressor pressure ratio, number of heat transfer units, and specific heat ratio on the cycle efficiency and power generation. The authors findings showed that the optimal specific heat ratio had increased as a result of the overall cycle pressure ratio decreasing, whereas a lower increase can be gained with higher intercooling pressure ratios. Al-Doori [25] studied the effect of different parameters, such as operation condition and design, on the power output, specific fuel consumption, compression work, and thermal efficiency of a gas turbine power plant with and without an intercooling system. The outcome of the research showed that the power generation efficiency of the power plant has improved by 9% from utilizing the intercooling system and that the performance of the intercooled gas turbine cycle can be further enhanced when both peak temperature and total pressure ratios are

increased. In addition, the intercooled cycle resulted in 8% less fuel consumption compared to the cycle operating without an intercooler unit.

### 3. Nanofluids Types and Fabrication Processes

As mentioned earlier, suspensions made of dispersed nanoparticles (NPs) in a non-dissolving liquid media is termed as nanofluids. The particles used in the fabrication of these advanced working fluids can be of metallic origin (e.g., aluminum (Al), copper (Cu), and iron (Fe)), metal oxides (e.g., $Al_2O_3$, CuO, and $Fe_2O_3$), carbon-based (e.g., carbon nanotubes (CNTs), graphene, and nanodiamond (ND)), alloys (e.g., different grades of stainless steels (SS)) . . . etc. [26]; whereas the common basefluids that are used to host the particles are water, oil, refrigerant, ethylene glycol (EG), methanol, or a combination of two or more of these liquids [2,27]. Furthermore, the term nanofluid can be further specified into two categories based on the dispersed particles, the first being the conventional nanofluid, where one type of particle is used in the production process; and the second is known as the hybrid nanofluids, which includes two or more sorts of dispersed nanomaterials in the basefluid [28]. Figure 5 shows an illustration of the two different categories of nanofluids.

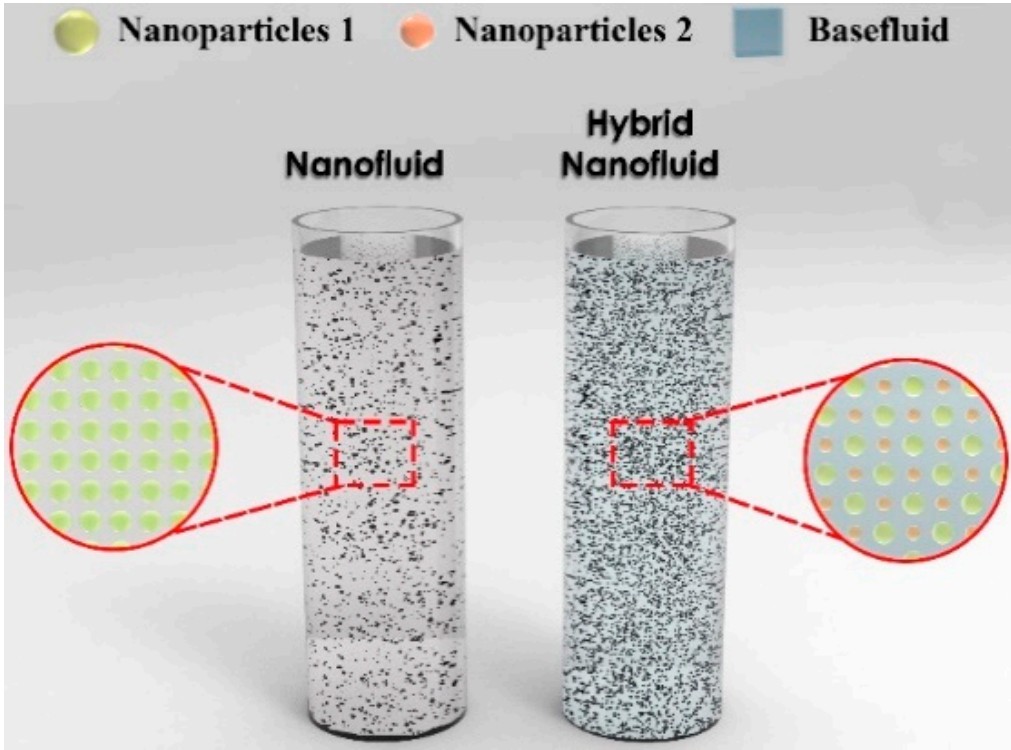

**Figure 5.** The two categories of nanofluids, where (**the left side**) exemplifies the conventional type of nanofluids, and (**the right side**) demonstrates the hybrid class of nanofluids.

There are two routes in which these advanced fluids can be fabricated, namely the single-step (or one-step) method, and the two-step approach [29,30]. In the one-step method the NPs are grown and dispersed instantaneously within the basefluid, while the two-step process relies on dispersing pre-prepared dried nanopowder in a host liquid through one or more mixing instruments. The advantages and disadvantages of each of the two previous fabrication methods are listed in Table 1 along with the devices (two-step approach) and conduction method (one-step process). A detailed explanation on each of the devices and methods of conduct that are listed in Table 1 can be found in the work of Mukherjee et al. [31] and Ali et al. [26]. In addition, to calculate the required volume percentage (vol. %) of particles to produce different combinations of nanofluids through the two-step approach, the reader is guide to the equations available in Ref. [2,26,32–35].

**Table 1.** Nanofluids production methods, advantages, disadvantages, and particles dispersion mean.

| Nanofluid Fabrication Methods | | |
| --- | --- | --- |
| **Production Route** | Single-step | Two-step |
| **Advantages** | • High dispersion stability.<br>• No drying, storage, and transportation of NPs is required. | • Any type of nanofluid can be produced.<br>• No residuals in the final product.<br>• Commercial NPs are widely available.<br>• Can easily be employed by manufacturers with average background.<br>• Appropriate for large- or small-scale production. |
| **Disadvantages** | • Residuals from reactants or uncompleted reactants are always present (i.e., impurity always exist in the final product).<br>• Limited by certain types of NPs and basefluids.<br>• Cannot be used for mass production. | • A preliminary nanomaterial synthesis stage is required.<br>• Producing a physically stable nanofluid is challenging.<br>• Process parameters (e.g., mixing duration, power intensity, and working temperature) are very crucial and highly effects the properties of the suspension. |
| **Particles Dispersion** | • Vacuum evaporation onto a running oil substrate (VEROS).<br>• Laser ablation.<br>• Microwave irradiation.<br>• Submerged arc nanoparticles synthesis system (SANSS).<br>• Phase transfer.<br>• Polyol method.<br>• Physical vapour condensation.<br>• Plasma discharge.<br>• Electrical explosion of wire. | • Homogenizer.<br>• Ultrasonic bath.<br>• Magnetic stirrer.<br>• Ball (or rod) milling. |

## 4. Dispersion Stability and Thermophysical Properties

The dispersion stability plays a major role in the resulting thermophysical properties of the as-prepared nanofluid, precisely the effective thermal conductivity and effective viscosity of the mixture. This is because, in an unstable state, the dispersed particles tend to draw each other into clustering, and thus forming larger sized solids (or agglomerations). As these agglomerated particles get heavier, they then separate from the basefluid as sediments due to the gravitational force [36]. Therefore, the effective thermal conductivity gets affected because: (1) the particle exposed surfaces have reduced, and (2) most of the liquid has lost its dispersed particles (in the fully unstable case). In addition, the effective viscosity would also increase since the new shear force required to move the overall fluid containing the agglomerated particles is much higher in such case. As for the effective density and effective specific heat capacity, these two properties are not affected by the dispersion stability because the first (i.e., density) is associated with the total mass and volume of the mixture, whereas the specific heat capacity is influenced by the NPs concentration that is added to the basefluid [37–39]. In other words, when increasing the NPs concentration, the heat transfer rate proportionally increases along its side, and hence causes the aforementioned thermal property to reduce. In general, the hosting basefluid has a higher specific heat capacity than the nanofluid produced from it [40,41]. One can accurately calculate the effective density by using the rule of mixture [42], whereas the effective specific heat can be determined through both the rule of mixture [42] and the thermal equilibrium model [43], although the second was proven to be more accurate than

the first [44]. Furthermore, the stability of the suspension can be determined after the fabrication stage through different approaches, such as the 3-ω method, zeta potential approach, dynamic light scattering analysis, sedimentation photographical capturing technique, scanning electron microscopy, etc. [26]. From these stability characterization methods, the sedimentation photographical capturing technique is considered as the most accurate tool for determining the NPs dispersion efficiency but is also very time consuming to conduct [36]. Nevertheless, it is very common to see the use of more than one of these models in any published work [45]. To overcome the physical stability obstacle, researchers have proposed the use of one or more of the following:

1. Selecting the appropriate combination of NPs and basefluid, based on their head group charges (e.g., metal oxides NPs with water or CNTs with oil) [46,47];
2. Employing physical methods, such as ultrasonication, magnetic stirring, homogenizer, or ball milling (Figure 6) [45]; and/or
3. Using chemical approaches, such as pH modification, surfactants (e.g., Arabic gum and Tween 80), and NPs surface functionalization [48,49].

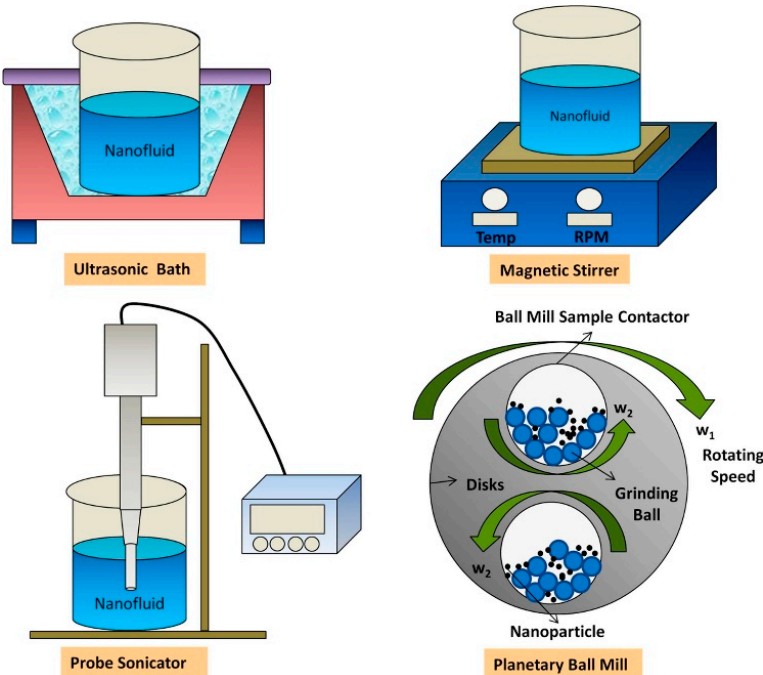

**Figure 6.** Different physical stability enhancement methods, where (**top left**) shows the ultrasonicator device, (**bottom left**) demonstrates the probe sonicator or homogenizer, (**top right**) illustrates the magnetic stirring apparatus, and (**bottom right**) shows the ball milling approach. Reproduced with permission from [45]. Elsevier, 2020.

At a stable and homogeneously dispersing state, the optimum nanofluid effective thermal conductivity (highest) and effective viscosity (lowest) conditions can be achieved. Some of the experimental measurement methods and predicting correlations for these two thermophysical properties are shown in Table 2. It is important to note that, up to today, all the effective thermal conductivity and effective viscosity formulas that are found in the literature are only suitable for the range of experiments from which they were developed.

**Table 2.** Selected effective thermal conductivity and effective viscosity measurement methods and theoretical formulas.

| Property | Method | Device/Equation * | Source |
|---|---|---|---|
| Effective thermal conductivity | Experimental | • Transient hot-wire.<br>• Cylindrical cell.<br>• Temperature oscillation.<br>• Steady state parallel-plate.<br>• 3-ω.<br>• Thermal constants analyzer.<br>• Flash lamp.<br>• Thermal comparator.<br>• Transient plane source.<br>• Laser flash. | [50–53] |
| | Formula * | $\dfrac{k_{eff}}{k_{bf}} = \dfrac{k_{pe}+2k_{bf}+2\left(k_{pe}-k_{bf}\right)\left(1-\frac{t_{nl}}{r_{np}}\right)^3 vol.}{k_{pe}+2k_{bf}-\left(k_{pe}-k_{bf}\right)\left(1-\frac{t_{nl}}{r_{np}}\right)^3 vol.}$, | [54] |
| | | $\dfrac{k_{eff}}{k_{bf}} = \dfrac{3+\eta^2 vol.}{\left[k_{bf}\left(\frac{2\,R_b}{L}+13.4\,\sqrt{t}\right)\right](3-\eta\ vol.)}$, | [55] |
| | | $\dfrac{k_{eff}}{k_{bf}} = \dfrac{1-vol.+2vol.\left(\frac{k_{np}}{k_{np}-k_{bf}}\right)\ \ln\left(\frac{k_{np}+k_{bf}}{2k_{bf}}\right)}{1-vol.+2vol.\left(\frac{k_{bf}}{k_{np}-k_{bf}}\right)\ \ln\left(\frac{k_{np}+k_{bf}}{2k_{bf}}\right)}$. | [56] |
| Effective viscosity | Experimental | • Rotational viscometer.<br>• Capillary viscometer.<br>• Concentric cylinder viscometer.<br>• Rheo-nuclear magnetic resonance.<br>• Rheo-Scope. | [57–60] |
| | Formula * | $\mu_{eff} = \mu_{bf}\ exp\left(\frac{FP._1}{T-FP._2}\right) + FP._3 vol.\ exp\left(\frac{FP._4}{T}\right) - FP._5 vol.^2$, | [61] |
| | | $\mu_{eff} = \mu_{bf}\left(-24.81 + 3.23\ T^{0.08014} exp\left(1.838 vol.^{0.002334}\right) - 0.0006779\ T^2 + 0.024\ vol.^3\right)$, | [62] |
| | | $\mu_{eff} = \mu_{bf}\left(1.15 + 1.061 vol. - 0.5442\ vol.^2 + 0.1181\ vol.^3\right)$. | [63] |

*—$k_{eff}$, $k_{bf}$, $k_{np}$, $k_{pe}$, $\mu_{eff}$, $\mu_{bf}$, $t_{nl}$, $r_{np}$, $R_b$, $L$, $\eta$, $t$, ($FP._{1-5}$), and $T$ are the effective thermal conductivity of the nanofluid, basefluid thermal conductivity, NPs thermal conductivity, particle equivalent thermal conductivity, effective viscosity, basefluid viscosity, nanolayer thickness, NP radius, impact of interfacial resistance, length of nanoplatelet, nanoplatelet thickness, average flatness ratio, temperature fitting parameters (available at the published source), and mixture temperature, respectively.

Moreover, scholars have published numerous amounts of research work that show the improvement in thermal conductivity of these suspensions over their base liquids [64–76]. Nevertheless, the inevitable increase in the effective viscosity caused by the dispersed particles in the basefluid was also commonly reported in literature [77,78]. Table 3 summarizes some of the investigations that were done on the effective thermal conductivity and effective viscosity of nanofluids.

**Table 3.** A summary of some of the available studies that focused on nanofluids effective thermal conductivity and viscosity.

| Researcher (/s) | NPs | Basefluid | Concentration | Additional Information | Finding |
|---|---|---|---|---|---|
| Yu et al. [64] | Graphene | EG | 2–5 vol. % | Two-step controlled temperature fabrication from 10 to 60 °C | The 5 vol. % suspension produced at 60 °C was 86% higher in thermal conductivity than pure EG at similar temperature. |
| Ghozatloo et al. [66] | - Graphene<br>- f-graphene | Water | 0.01–0.05 wt. % | - An alkaline additive was used in the functionalization process.<br>- Two-step process took 1 h. | - f-graphene showed high physical stability<br>- 0.05 wt. % and 0.03 wt. % f-suspensions had 13.5% and 17% higher thermal conductivity than water at 25 and 50 °C, respectively. |
| Zhang et al. [67] | - Graphite<br>- Graphene<br>- CNTs | Ionic liquid | – | Dispersion of particles was done through two stages. The first was through magnetic stirring (15 min), and the second by ultrasonication (1 h). | - All suspensions showed higher thermal conductivity than their basefluids.<br>- Graphene had the highest improvement over the other types of NPs. |
| Timofeeva et al. [75] | Silicon carbide | EG–water | – | - NPs were produced from Saint Gobain Inc.<br>- Particles size: 16–90 nm.<br>- Basefluid pH: 9.5 ± 0.3.<br>- Thermal conductivity of the nanofluids were measured more than 100 time (15 min between each) then averaged. | - All as-prepared nanofluids had higher thermal conductivity than their basefluids.<br>- Increasing the NPs size caused the thermal conductivity of the suspension to increase.<br>- Highest reported heat transfer enhancement was 14.2%. |
| Timofeeva et al. [76] | f-graphite nanoplatelets | EG–water | Up to 5 wt. % | - NPs thickness: 2–12 nm.<br>- NPs diameter: 0.1–25 μm.<br>- Functionalization was done using $H_2SO_4$ and $HNO_3$ of 3:1 ratio. | Increasing the NPs thickness and diameter have caused the effective thermal conductivity to improve up to 80% over the basefluid but had also raised the effective viscosity 100 times more. |
| Akhavan-Zanjani et al. [77] | Graphene | Water | 0.005–0.02 wt. % | - Polyvinyl alcohol was used as surfactant.<br>- Two-step sonication method was used. | 0.02 wt. % nanofluid showed an increase of 4.95% in viscosity compared to the basefluid. |
| Li et al. [78] | Silicon dioxide | Liquid paraffin–oleic acid | 0.005–5 wt. % | - Two-step controlled temperature approach.<br>- Mixing temperatures were from 25 to 70 °C. | significant increases in the 5 wt. % nanofluids viscosity over the basefluid, precisely 331% (at 25 °C) and 495% (at 70 °C). |

Note: f-graphene and wt. % referee to functionalized graphene and weight percentage, respectively.

## 5. Utilization of Nanofluids in Intercoolers

In Section 4, it was demonstrated how nanofluids are favorable in terms of thermal transport compared to their conventional counterparts. The effective thermal conductivity of these advanced fluids makes them promising candidates for most (if not all) heat transfer applications that utilize liquids as working fluids. It was also shown that these suspensions have two main drawbacks, i.e., maintaining the physical stability of the dispersion and an increase in the effective viscosity caused by the added NPs. However, these limitations can be resolved at the nanofluids fabrication stage. Their effect on the thermal performance of any hosting system depends on many aspects, such as the type of nanomaterials used, type of basefluid, level of physical stability, NPs concentration, flow rate, system geometrical design, working temperature, etc. [2]. Furthermore, when it comes to investigating the employment of these types of fluids in intercoolers, there are very few published articles that have covered this topic [79–83].

Zhao et al. [79,80] theoretically studied the effectiveness of replacing conventional coolants with $Al_2O_3$–water and Cu–water nanofluids in a marine gas turbine intercooler system. The authors used the effectiveness-number of transfer unit approach to estimate both flow and thermal transport performance of the heat exchange unit. The NPs concentration for the two types of nanofluids were between 1 to 5 vol. % and the suspensions thermophysical properties were calculated using pre-existing correlations from the literature. The influence of key parameters, such as NPs concentration, nanofluids inlet temperature, Reynolds number (*Re*), and other parameters concerned with the gaseous side of the intercooler were explored in terms of heat transfer and flow performance of the thermal transport device. The outcome of their research showed that the nanofluids have surpassed the heat transfer performance of the conventional coolant (i.e., water), while requiring less pumping power at all gas turbine operating scenarios. On the other hand, Zhao et al. [84] explored theoretically the effect of the intercooler material and operational condition, of a similar gas turbine engine as the one studied by Zhao et al. [79,80], on the working fluid. The authors investigated three intercoolers materials, namely copper-nickel alloy, aluminum, and copper. They found that placing an intercooler of copper-nickel alloy and increasing the operational load from ~20% to 100% caused the outlet temperature of the compressed gas to rise from ~26.4 to ~39.4 °C and from ~26.7 to ~40.2 °C when using constant properties and variable properties, respectively. Furthermore, the coolant temperature was also reported to elevate, from ~23.3 to ~30.3 °C, with the increase in operational load (i.e., from 20% to 100%), but the influence of using variable or constant working fluid properties in their simulation have shown negligible variation in the results. The aforementioned finding illustrates that the compressed gas parameters are more sensitive to their working condition in comparison to the coolant. In terms of intercooler material, they found that the dynamic response of the compressed gas was the highest when the HE device was made of copper-nickel alloy, whereas the dynamic response of the coolant had a better performance with the aluminum-based intercooler. In addition, the aluminum HE demonstrated a better dynamic response balance between the two working fluids (i.e., compressed gas and coolant) when compared to the other two alternative options. Another study by Masoud Hosseini et al. [81] investigated the performance of the intercooler of a liquefied petroleum gas (LPG) absorber tower when utilizing carbon-based nanofluids of low concentrations. The suspensions were made of 0.0055–0.278 vol. % of CNTs dispersed in water, whereas the type of the intercooler was of shell and tube. Furthermore, the design, simulation, and evaluation of the HE device performance was conducted using commercial software called ASPEN HTFS + 7.3. The researchers found that the 0.278 vol. % working fluid caused the heat transfer rate and overall heat transfer coefficient to improve over pure water by 10.3% and 14.5%, respectively. They indicated that their findings can be beneficial for manufacturers in the sense that the heat transfer area of the heat exchanger can be reduced, and this would consequently lower the net production cost of the thermal device. Estellé [82] showed his interest in Masoud Hosseini et al. [81]'s published work because it reflects the utilization of nanofluids in an actual industrial application but also argued that the authors [81] should have used a more suitable effective viscosity formula, precisely one that is developed for CNTs and not for spherical $Al_2O_3$ particles. He further stated that the accuracy of

parameters such as the pressure drop, *Re*, and heat transfer coefficient would only be affected when modeling nanofluids contained 0.111 vol. % or higher of CNTs. Chintala et al. [83] experimentally investigated the heat transfer behavior of a two-stage air compressor intercooler using $Al_2O_3$–water suspensions. Their nanofluids were fabricated by dispersing 0.5 to 1 vol. % of $Al_2O_3$ at 40 °C and for 90 min, using a probe sonicator. Moreover, the studied intercooler had a counterflow arrangement. Their results showed that the overall heat transfer coefficient and effective thermal conductivity of the as-prepared suspensions increased with the increase of NPs concentration. The highest attained thermal conductivity was 0.677 W/m.K (for the 1 vol. % nanofluid) and the lowest was ~0.6 W/m.K (for the pure basefluid). In addition, the efficiency of the intercooler showed a 36.1% enhancement with the 1 vol. % suspension, at a compressor load of 4 bar.

All the encouraging findings from the literature that were covered gives an indication that utilizing nanofluids in intercoolers would be of great advantage to the industry. However, the number of publications are not sufficient at this stage to fully support this idea. A summary of the covered published work is listed in Table 4.

Table 4. Summary of the published work that was covered in Section 5.

| Scholar (/s) | NPs Type | Basefluid | NPs Concentration (vol. %) | Performance Enhancement | Additional Notes |
|---|---|---|---|---|---|
| Zhao et al. [79,80] | • $Al_2O_3$ <br> • Cu | Water | 1–5 | Pumping power <br> • 2.08% ($Al_2O_3$) <br> • 6.42% (Cu) | The study was performed theoretically on a marine gas turbine system that is integrated with an intercooler HE. |
| Masoud Hosseini et al. [81] | • CNTs | Water | 0.0055–0.278 | Heat transfer rate 10.3% | Theoretical modelling of an industrial LPG absorber tower utilizing CNTs nanofluids. |
| Estellé [82] | • CNTs | Water | 0.0055–0.278 | – | The author commented on the previous work of Masoud Hosseini et al. [81], where he pointed out that the effective viscosity equation used in the modeling process was inappropriate for the task. This is because the researchers used the formula for dispersed $Al_2O_3$ NPs instead of CNTs, which will only affect the results of nanofluids with ≥0.111 vol. % |
| Chintala et al. [83] | • $Al_2O_3$ | Water | 0.5–1 | Intercooler efficiency 36.1% at 4 bar compressor load | Experimental investigation on a double stage air compressor fitted with an intercooler HE. |

## 6. Discussion and Future Directions

The variation in intercooler designs and flow arrangements was demonstrated for different real-life gas turbine systems. It was shown how nanofluids could be of great benefit for enhancing the thermal performance of these thermal devices. Furthermore, the description of the two categories of nanofluids (i.e., hybrid-nanofluids and conventional nanofluids) were provided as well as the theoretical calculation of the volumetric concentration of NPs needed to form these type of suspensions. Moreover, the dispersion stability of nanofluids was shown to strongly influence their thermophysical properties, precisely the effective thermal conductivity and effective viscosity, but such negative effect can be reduced through different mechanical and chemical stability enhancement approaches (e.g., sonication or adding surfactants). In addition, the previous investigations that were covered in this review have proven that replacing conventional fluids with nanofluids would cause the thermal performance of the hosting system to greatly improve. Nevertheless, there are still many issues that needs to be considered before such advanced types of fluid can be commercially accepted, especially for gas turbine intercoolers. Some of these issues are listed below:

- A feasibility study on utilizing different types of nanofluids needs to be conducted. It should consider the net cost of the starting materials (e.g., NPs and basefluids) against the gained performance enhancement of the system. At this point, there does not exist any such study on intercooler HEs [85].
- One of the most important aspects that is associated with the use of nanofluids is their environmental impact. This includes the different stages that the nanofluids undergo, such as NPs synthesis, nanofluids production, suspension transfer and employment in the system, and disposal. Currently, such studies are very limited, and at the same time, not available for intercoolers [86].
- Further investigation in terms of both experimental and theoretical approach should be conducted on the effect of employing nanofluids in gas turbine intercooler systems. This is because the current literature is not adequate for introducing such heat transfer fluids for the industry.
- In terms of dispersion stability, the chemical routes used for improving the stability of nanofluids, such as surfactants and NPs functionalization, would also cause their effective thermal conductivity to reduce [48,87]. The level of degradation in such thermal property is crucial when it comes to their thermal transport capability, and hence more investigations are needed in this area.
- Generally, in any application of elevated temperature, the working fluid tends to form fouling layers on the attached surfaces. Such layer formation is more rapidly seen when using nanofluids except for the thin film in such case would mostly contain deposited NPs that were initially hosted by the working fluid [2]. Researchers [88–90] have reported that the wettability of the surface changes with the type of film formed, layer thickness, and temperature of the attached fluid. Understanding how nanofluids fouling affect the wettability of different surfaces is important as it is directly linked to the pumping power required by the system, and hence the overall efficiency of the system.

## 7. Conclusions

This work has provided a short review of the available literature on gas turbine intercoolers and nanofluids as their working fluids. The survey started by including the different gas turbine engines, intercoolers designs, flows types, and arrangements. It was shown that gas turbine intercoolers can be designed as shell and tube or plate-fin heat exchangers, whereas their flow arrangements can be of counterflow, parallel flow, or crossflow. Furthermore, the improvements that the intercooler delivers when coupled to a gas turbine cycle was demonstrated through the findings of published works in the field. In addition, the conducted review had also covered the different types of nanofluids, their fabrication approaches, dispersion physical stability, and the thermophysical properties of these suspensions. The thermal property of these suspensions was shown to be significantly higher than conventional working fluids, which therefore makes them favorable for heat transfer applications.

The downside from using nanofluids was that forming a physically stable dispersion may be challenging and that they always have a higher viscosity than their basefluids. This could be problematic in real-life applications but can be resolved at the fabrication stage of these advanced fluids. Available studies on the use of nanofluids in intercoolers have shown promising system performance enhancement, which gives a good indication for the future of these types of working fluids.

**Author Contributions:** A.A. and N.A. conducted the Introduction Section. A.A. carried out the Gas Turbine Intercooled System and Their Design Section. A.A. and N.A. conducted the Nanofluids Types and Fabrication Processes Section. A.A. and N.A. worked on the Dispersion Stability and Thermophysical Properties Section. A.A. conducted the Utilization of Nanofluids in Intercoolers Section. A.A. and N.A. have worked on the Discussion and Future Directions Section along with the Conclusion Section. All authors have read and agreed to the published version of the manuscript.

**Funding:** This research received no external funding.

**Acknowledgments:** The authors of this article are grateful and acknowledge the help provided by their institutes.

**Conflicts of Interest:** The authors declare no conflict of interest.

## Nomenclature

| | |
|---|---|
| CNT | Carbon nanotube |
| EG | Ethylene glycol |
| *FP.* | Temperature fitting parameter |
| HE | Heat exchanger |
| *k* | Thermal conductivity (W/m.K) |
| *L* | Length of nanoplatelet (nm) |
| ND | Nanodiamond |
| NP | Nanoparticle |
| P&FHE | Plate-and-frame heat exchanger |
| PFHE | Plate-fin heat exchanger |
| PHE | Plate heat exchanger |
| *r* | Radius (nm) |
| $R_b$ | Impact of interfacial resistance |
| *Re* | Reynolds number |
| SANSS | Submerged arc nanoparticles synthesis system |
| SS | Stainless steel |
| *T* | Mixture temperature (°C or K) |
| *t* | Average flatness ratio |
| $t_{nl}$ | Nanolayer Thickness (nm) |
| VEROS | Vacuum evaporation onto a running oil substrate |
| vol. % | Scanning electron microscope |
| wt. % | Weight percentage |
| Greek letters | |
| $\eta$ | Nanoplatelet thickness (nm) |
| μ | Dynamic viscosity (kg/m.s) |
| Subscripts | |
| *bf* | Basefluid |
| *eff* | Effective |
| *np* | Nanoparticles |
| *pe* | Particle equivalent |

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
