# Peer review of "Gas Turbine Intercoolers: Introducing Nanofluids—A Mini-Review"

_processes, doi:10.3390/pr8121572_

Round 1

Reviewer 1 Report

This short review is scoped to cover heat transfer enhancement in gas turbine inter coolers using nanofluid. Overall, this is a good review, however, I would recommend the authors to have a more comprehensive review, I.e., not restricting themselves only thru Scopus database. I would recommend its publication.

additional comments:

This review article aims to review state of the art for heat enhancement using nanofluid in gas turbine intercoolers. The topic itself is relevant to processes and interesting to both academia and industry. There has been quite a few review papers on nanofluid heat transfer enhancement in scientific community, however, this article is focused on gas turbine intercoolers specifically, which looks original from the scope point of view. However, this review article mostly reports general aspects of nanofluid: thermal conductivity enhancement, viscosity, suspension stability, fabrication, etc. It is still short of specific review on applications of using nanofluid on gas turbine intercoolers specifically. The authors should do a more comprehensive review specific to gas turbine intercoolers as the title indicates, for example, pros and cons, academia achievement, industry use, etc. As for references, for example, recently Argonne National Lab has been working with Saint Gobain to produce SiC in EG/water; Valvoline to produce graphite-based nanofluid, etc. which may look interesting to readers from an industry practical use point of view.

Author Response

We thank the respected reviewer for his time and support. We also thank him for helping us improve our manuscript through the comments that were provided. Kindly note that all the comments were taken into account in our revised version of the manuscript and can be seen highlighted in the text. The following have been addressed:

Reviewer: I would recommend the authors to have a more comprehensive review, I.e., not restricting themselves only thru Scopus database.

Authors: Two research databases were added in the analysis of the available literature. This can be seen in Lines 54 – 69. In addition, a compression between the all three databases was performed, as shown in Figure 1.

Reviewer: The authors should do a more comprehensive review specific to gas turbine intercoolers as the title indicates, for example, pros and cons, academia achievement, industry use, etc..

Authors: The respected reviewer comment was taken into consideration (please see lines 97 – 103). In addition, the authors have added some of the studies available in the literature that shows the important role of intercoolers in improving the gas turbine cycle performance (please see lines 132 – 150).

Reviewer: As for references, for example, recently Argonne National Lab has been working with Saint Gobain to produce SiC in EG/water; Valvoline to produce graphite-based nanofluid, etc. which may look interesting to readers from an industry practical use point of view.

Authors: We totally agree with the respected reviewer comment. Kindly note that the authors have included the recommended work along with others (i.e., Ref. 69 – 74) that were found very useful for enriching the manuscript content (please see lines 285 – 304).  

Thank you very much for your kind help and support.

Best regards,

The Authors

Reviewer 2 Report

I read this review paper. These are my comments on your manuscript.

  • Review papers on Nanofluids are published in many in decay. This is not new. Also, fabrication methods and processes are not impactive features for this area at this moment.
  • A large portion of your review work is described with basic history or preliminary knowledge of nanofluids.
  • Your concern on application to intercooler should be taken attention on your presentation.
  • Your main portion of the manuscript will be Section 5. The other parts of your paper are not necessary. I recommend that your review work should be focused on the intercooler with nanofluids.
  • I can not catch what you want to claim or present in your study.
  • Your conclusion can not transfer any information on intercooler with nanofluid. You should improve your conclusion based on review work.

Author Response

We thank the respected reviewer for his time and comments. Kindly note that the following:

Reviewer Extensive editing of English language and style required.

Authors: Kindly note that the English language of the manuscript has been revised and improved, as can be seen in the revised version of the manuscript.

Reviewer: Review papers on Nanofluids are published in many in decay. This is not new.

Authors: Kindly note that review papers on nanofluids are still been published in many highly respected journals (e.g., MDPI Nanomaterials, Renewable and Sustainable Energy Reviews, Solar Energy, International Journal of Refrigeration … etc.), which are all in 2020. If you kindly check the scopus database, you will find that 100 review articles have been published on nanofluids in 2020 only. This shows how important this topic is to the research society. We would also like to point out that the in-hand review article is the first and only review work ever done on nanofluids for gas turbine intercoolers, which therefore is expected to have a great impact on the field.

Reviewer: Also, fabrication methods and processes are not impactive features for this area at this moment.

Authors: We thank the respected reviewer for his comment but would like to point out that the fabrication stage of any suspension (i.e., not just nanofluids) has a major impact on the resulting thermophysical property. This is because it effects the dispersed particles stability, which therefore causes the resulting thermophysical properties to greatly varies. As such, if not fully understood and handled, the user will not be able to utilize the full potential of the desired suspension property. Therefore, it will definitely have a great impact on the in-hand application (i.e., intercooler).

Reviewer: A large portion of your review work is described with basic history or preliminary knowledge of nanofluids.

Authors: We thank the respected reviewer for his comment. Kindly note that the majority (if not all) review articles on nanofluid tend to cover the main aspects of such type of advanced fluids (e.g., https://doi.org/10.1016/j.ijheatmasstransfer.2020.119611 and https://doi.org/10.1016/j.rser.2018.12.057). In general, it is very important to provide such knowledge to the respected readers, especially in our case, where most of them would have gas turbine backgrounds with very few (if any) knowledge on nanofluids. Therefore, we are trying to show these groups of scientist and other interested researchers the important role that nanofluids can deliver when employed as working fluids in gas turbine intercoolers. This would also help promote the research on these type of advance fluids for gas turbine applications, and if proven feasible could possibly be tested on real life scale engines by major research groups, such as ANL.    

Reviewer: Your concern on application to intercooler should be taken attention on your presentation.

Authors: We thank the respected reviewer for his comment. Kindly note that we have improved our Gas Turbine Intercooled System and Their Designs Section (Please see Section 2, lines 96 – 103 and 132 - 150), as recommended.

Reviewer: Your main portion of the manuscript will be Section 5. The other parts of your paper are not necessary. I recommend that your review work should be focused on the intercooler with nanofluids.

Authors: We thank the respected reviewer for his comment. Kindly note that the manuscript was designed to cover the targeted application, the working fluid (i.e., nanofluid) then their combination. This systematic approach was adopted so that the reader can follow as to why nanofluids are important for gas turbine intercooler. Kindly also not that in our title we mentioned that this is a Mini-Review. The reason behind the selected term is that at the current state, there isn’t many published work on the utilization of nanofluids in gas turbine intercoolers.   

Reviewer: I can not catch what you want to claim or present in your study.

Authors: We apologies for that and hope that from our previous comments replies and modified version of our manuscript the picture can be more clearer.

Reviewer: Your conclusion can not transfer any information on intercooler with nanofluid. You should improve your conclusion based on review work.

Authors: We thank the respected reviewer for his comment. Please note that in our conclusion we provide general outcomes on the part that you have mentioned (please see lines 422 – 424), whereas the precise details can be found in the body of the manuscript. In general, there are different ways in writing the Conclusion Section of a review article. For instant, you may find some review articles having the future work in their conclusion section, and vice versa. Others prefer to be very detailed in their conclusion section. Despite that, they are all correct ways of writing a conclusion section.

Finally, we hope that we were able to address your comments to an acceptable standard and do apologies if we couldn’t.

We thank you very much for accepting to review our work and for sharing with us your valuable time.

Best regards,

The Authors

Reviewer 3 Report

The authors have presented a short review on the effect of different types of nanofluids on the thermal performance of gas turbines intercooler. They also mentioned the current changes and future research directions on the performance of the intercooler using the nanofluids. This review study is informative and interesting.  I am recommending this paper for publication with minor revision.

Minor issues are given below:

Page – 1

Abstract

Line 9: 1st word “coolant” should be capitalized to “Coolant”.

Line 15: 8th and 9th word “types of” should be deleted.

Line 19: 3rd word “highlight” should be changed to “highlighted”.

  1. Introduction

Line 26: 1st word “continues” should be changed to “continuous”.

Line 28: 9th word “current” should be changed to “currently”.

Line 31: 7th word “year” should be changed to “years”.

Line 32: 2nd last word “reheat” should be changed to “reheated”.

Line 35: 6th last word “rise” should be changed to “raise”.

Line 40: 9th word “in” should be changed to “for”.

Line 41: 1st word “for” should be deleted.

Line 42: 4th word “equipment’s” should be changed to “equipment”.

Page – 2

Line 44: 8th word “into” should be deleted.

Line 49: 6th word “in” should be changed to “to”.

Line 53: 8th word “nanofluids” should be changed to “ nanofluids’ ”.

Line 55: 5th last word “contained” should be changed to “contain”.

Line 64: 2nd last word “searching” should be changed to “search”.

Line 65: 5th and 6th words “in hand” should be combined to “in-hand”.

Line 67: 2nd word “authors” should be changed to “ authors’ ”.

Line 68: “the” should be added after the 5th word “presenting”.

Line 70: 4th word “nanofluids” should be changed to “ nanofluid’s ”.

Line 71: 4th last word “summarises” should be changed to “summarizes”.

Line 72: 4th word “intercoolers” should be changed to “ intercoolers’ ”.

  1. Gas Turbines Intercooled System and Their Designs

Line 78: 9th word “continues” should be changed to “continuous”.

Line 81: 3rd word “electrical” should be changed to “electric”.

Line 85: 2nd and 3rd last word “aero engines” should be combined to “aero-engines”.

Line 86: 2nd last word “marines” should be changed to “marine”.

Page – 3

Line 95: 6th word “passes” should be changed to “pass”.

Line 97: 7th word “inters” should be changed to “enter”.

Line 99: 1st word “fluids” should be changed to “fluid”.

Page – 4

Line 106: 6th last word “meets” should be changed to “meet”.

                4th last word “requirement” should be changed to “requirements”.

Line 107: 4th last word “types” should be changed to “type”.

Page – 5

  1. Nanofluids Types and Fabrication Processes

Line 116: 8th word “aluminium” should be changed to “aluminum”.

Line 122: 6th word “particles” should be changed to “particle”.

Page – 6

Line 128: 3rd last word “through” should be deleted.

Line 129: comma should be added after the 4th last word “method”.

Line 138: “the” should be added after 5th last word “within”.

Line 139: 2nd word “synthesised” should be changed to “synthesized”.

Line 140: 6th word “approaches” should be changed to “approach”.

Page – 7

Page – 8

  1. Dispersion Stability and Thermophysical Properties

Line 171: comma should be added after the 4th word “because”.

Line 173: 1st word “gets” should be changed to “get”.

Line 173: comma after the 6th last word “sediments” should be deleted.

Line 174: 3rd last word “particles” should be changed to “particle”.

Line 179: 9th word “by” should be changed to “with”.

Line 181: 3rd word “concept” should be changed to “other words”.

Line 183: “a” should be added after the 7th word “has”.

Line 187: 4th word “accuracy” should be changed to “accurate”.

Page – 9

Line 201-202: The sentence “Nevertheless, it is very common to see in any published work more than one of these methods employed” should be rephrased as “Nevertheless, it is very common to see the use of more than one of these models in any published work”.

Line 203: comma should be added after the 3rd word “obstacle”.

Line 219: 4th word “in” should be changed to “from”.

                 Last word “from” should be deleted.

Page – 10

Page – 11

Line 232: 1st word “there” should be changed to “their”.

Line 234: 7th word “an” should be deleted.

Line 236: Last word “nonfunctionalized” should be changed tonon-functionalized”.

Line 241: 7th word “where” should be changed to “were”.

Line 248: 4th word “conventional” should be changed to “conventionally”.

                 4th last word “there” should be changed to “the”.

Line 252: Last two words “poly vinyl” should be combined to polyvinyl”.

Line 256: 8th and 9th words “on the” should be deleted.

  1. Utilization of Nanofluids in Intercoolers

Line 262: 10th word “favourable” should be changed to favorable”.

Line 264: 2nd word “makes” should be changed to “make”.

Line 266: 9th word “the” should be changed to “an”.

Line 267: 10th word “overcome” should be replaced by “resolved”

Line 274: 3rd word “wate” should be changed to “water”.

Page – 12

Line 276: Last word “where” should be changed to “were”.

Line 277: 3rd last word “where” should be changed to “were”.

Line 280: 3rd word “gas” should be changed to “gaseous”.

                 comma after the 7th word “intercooler” should be deleted.

                 8th word “where” should be changed to “were”.

Line 284: 1st word “has” should be changed to “have”.

Line 285: 3rd last word “where” should be changed to “were”.

Line 294: 6th last word “have” should be changed to “has”.

Line 297: Last word “modelling” should be changed to “modeling”.

Line 299: 4th word “behaviour” should be changed to “behavior”.

Line 303: 4th last word “in” should be changed to “of”.

Line 309: 3rd last word “summery” should be changed to “summary”.

Line 310: 4th word “is” should be changed to “are”.

Page – 13

  1. Discussion and Future Directions

Line 314: 4th word “intercoolers” should be changed to “intercooler”.

Line 317:  6th last word “was” should be changed to “were”.

Line 318: 2nd last word “types” should be changed to “type”.

Line 324: “are” should be added after the 3rd last word “there”.

Line 325: 4th last word “fluids” should be changed to “fluid”.

Line 326: 1st word “acceptable” should be changed to “accepted”.

Line 329: comma should be added after the 8th word “point”.

Line 326-327:  The sentence “More investigations, both experimental and theoretical, on the effect of employing nanofluids in gas turbine intercooler system are needed.” should be rephrased as “Further investigation in terms of both experimental and theoretical approach should be conducted on the effect of employing nanofluids in gas turbine intercooler systems”.

Page – 14

Line 340: 3rd last word “causes” should be changed to “cause”.

Line 346: “for” should be added after the 2nd word “except”.

Line 349: 2nd last word “effect” should be changed to “affect”.

  1. Conclusion

Line 353: 8th word “on” should be changed to “of”.

Line 359: 4th word “theses” should be changed to “these”.

Line 360: 3rd and 4th last word “there for” should be combined to “therefore”

Line 361: 1st word “favourable” should be changed to “favorable”.

Line 362: 2nd word “stabile” should be changed to “stable”.

                 “a” should be added after the 4th last word “have”.

Author Response

We thank the respected reviewer for his time and support. We also thank him for helping us improve our manuscript through the provided comments. Kindly note that all the comments were taken into account in our revised version of the manuscript, as highlighted in the text.

Thank you very much sir for your kind help and support.

Best regards,

The Authors

Round 2

Reviewer 2 Report

I read your revised paper. But it is not enough qualifying your scope for reviewing intercooler & nanofluid.  Your review paper is too much orientated to nanofluid. 

Author Response

We thank the respected reviewer for sharing with us his valuable time and for accepting to review the revised version of our manuscript. We also appreciate his fruitful comments that helped us improve our manuscript. The respected reviewer remarks will surely help us develop our research skills and writing capability in the future.

Thank you very much sir.

The Authors